# Health Care Workers’ Perspectives of the Influences of Disrespectful Maternity Care in Rural Kenya

**DOI:** 10.3390/ijerph17218218

**Published:** 2020-11-06

**Authors:** Adelaide Lusambili, Stefania Wisofschi, Constance Shumba, Jerim Obure, Kennedy Mulama, Lucy Nyaga, Terrance J. Wade, Marleen Temmerman

**Affiliations:** 1Department of Population Health (DPH), Aga Khan University, Nairobi P.O. Box 30270-00100, Kenya; constance.shumba@aku.edu; 2Centre of Excellence in Women and Child Health, Aga Khan University, Nairobi P.O. Box 30270-00100, Kenya; swisofschi@gmail.com (S.W.); jerobure2000@gmail.com (J.O.); kenmulama@gmail.com (K.M.); lucy.nyaga@aku.edu (L.N.); twade@brocku.ca (T.J.W.); marleen.temmerman@aku.edu (M.T.); 3Department of Health Sciences, Brock University, St. Catharines, ON L2S 3A1, Canada

**Keywords:** disrespectful maternity care, respectful care, violence, maternity, obstetric, rural, Kenya

## Abstract

While disrespectful treatment of pregnant women attending health care facilities occurs globally, it is more prevalent in low-resource countries. In Kenya, a large body of research studied disrespectful maternity care (DMC) from the perspective of the service users. This paper examines the perspective of health care workers (HCWs) on factors that influence DMC experienced by pregnant women at health care facilities in rural Kisii and Kilifi counties in Kenya. We conducted 24 in-depth interviews with health care workers (HCWs) in these two sites. Data were analyzed deductively and inductively using NVIVO 12. Findings from HCWs reflective narratives identified four areas connected to the delivery of disrespectful care, including poor infrastructure, understaffing, service users’ sociocultural beliefs, and health care workers’ attitudes toward marginalized women. Investments are needed to address health system influences on DMC, including poor health infrastructure and understaffing. Additionally, it is important to reduce cultural barriers through training on HCWs’ interpersonal communication skills. Further, strategies are needed to affect positive behavior changes among HCWs directed at addressing the stigma and discrimination of pregnant women due to socioeconomic standing. To develop evidence-informed strategies to address DMC, a holistic understanding of the factors associated with pregnant women’s poor experiences of facility-based maternity care is needed. This may best be achieved through an intersectional approach to address DMC by identifying systemic, cultural, and socioeconomic inequities, as well as the structural and policy features that contribute and determine peoples’ behaviors and choices.

## 1. Introduction

Disrespectful maternity care (DMC) is a global issue affecting how mothers are treated during prenatal, delivery, and postnatal care. However, in low- and middle-income countries (LMICs) such as Kenya, these experiences are known to discourage or delay maternal care in health facilities [1,2,3,4,5], leading to higher rates of maternal and neonatal morbidity and mortality [6,7,8]. The World Health Organization defines respectful maternity care (RMC) as “care organized for and provided to all women in a manner that maintains their dignity, privacy and confidentiality, ensures freedom from harm and mistreatment, and enables informed choice and continued support during labor and childbirth” [9]. Respectful maternity care emphasizes that quality of care is achieved through the provision of high-quality, evidence-based, and informed services, practices, and care while recognizing the unique needs and preferences of women and newborns [10,11]. In contrast, disrespectful maternity care is not only the absence of respectful care that can threaten the dignity, privacy, and confidentiality of mothers but, also, causes psychological or physical harm to the mother through discrimination and maltreatment, obstruct or prevent choice in treatment, and fail to provide support to the mother and baby during labor and childbirth. The adoption and contextualization of standards of respectful care and quality measures are important to encourage facility-based care among mothers, leading to better birthing experiences and outcomes for mothers and newborns [9].

Most research on the disrespectful treatment of pregnant women during facility care, both in Kenya and beyond, is from the perspective of the service users, including mothers and family members [1,2,12,13]. In Figure 1 below, we summarize these findings from research across sub-Saharan Africa (SSA) on the influences of disrespectful experiences of pregnant women during facility care. Findings primarily from service users’ perspectives in Figure 1 [1] indicate that staff attitudes could be influenced by the pregnant women’s age (being young or old), being poor, and other culturally based considerations, such as religion and ethnicity. As shown in Figure 1, the interface between age, disability, religion, socioeconomic status, and social and cultural norms may increase the barriers experienced by women in accessing care at the facility level [1].

Research from high-income countries shows that HCWs sometimes show disrespectful maternity care particularly to women from low social economic status, such as migrants and refugees [14,15,16,17,18]. Poor communication between service users as a result of language barriers and illiteracy among ethnic minority women may contribute to negative attitudes towards them by HCWs [14,15,16,17,18]. While this is also true in sub-Saharan Africa (SSA), emerging research from the perspective of health care service providers also connects the poor treatment of women by health care workers (HCWs) to health systems constraints [19]. For example, a qualitative study among HCWs in Nigeria found that the uncooperative behavior of some women during labor has been reported to trigger attending staff to verbally abuse them [19]. Fear of stigma and discrimination—for example, toward the age or wealth of the mother—has been reported by HCWs as an important factor that make women less likely to use health facilities in LMICs during facility-based delivery [1,2,20]. As well, health care workers may be aware of the need to deliver respectful maternity care but fail to practice or implement it due to context-based barriers. For example, a study among midwives in Ghana found that while midwives demonstrated some awareness of RMC, there was a gap between knowledge and practice attributed to motivational, institutional, and sociocultural barriers [21]. Given that obstetric care by skilled providers is recognized as essential to reducing maternal and neonatal mortality, the practice of disrespectful care that discourages facility-based deliveries has the potential to undermine health system developments and investments made in Kenya toward advancing maternal health outcomes.

Aga Khan University has been conducting programmatic work in Kilifi and Kisii, Kenya on maternal and new child health since 2015 with financial support from the Government of Canada through Global Affairs Canada (GAC) and the Aga Khan Foundation Canada (AKFC). As part of the program, we conducted research [1] that revealed service users’ experiences of disrespectful maternity care during antenatal, delivery, and postnatal care by HCWs. However, there are gaps in our understanding as to why HCWs deliver DMCs to pregnant mothers in rural Kilifi and Kisii. We conducted the current study with the aim of understanding HCWs’ perspectives of RMC in the same setting in order to fill in unaddressed gaps from our previous research [1].

## 2. Methods

### 2.1. Study Context

A descriptive qualitative study involving in-depth interviews with 24 HCWs was conducted. Interviews took place between January and March 2020 in rural Kilifi and Kisii counties, where the Aga Khan University has been implementing a Maternal Newborn and Child Health (MNCH) project since 2015. Kilifi and Kisii are two of the poorest counties in Kenya, with high maternal mortality and morbidity rates. In Kenya, the proportion of women delivering at health facilities is 61%, while, in Kilifi and Kisii, this stands at 52.6% and 69%, respectively [22]. The poor treatment of women by HCWs during pregnancy and delivery in part contributes to these relatively low figures [13,18]. Many women continue to deliver at home with the help of traditional birth attendants (TBAs). Further, the rate of teenage pregnancies in both counties remains higher than the national average [22].

### 2.2. Study Sample Participants

A total of 24 HCWs (18 females and 6 males), 12 in each site, who worked for at least one year in Access to Quality Care for Extending and Strengthening Health Services (AQCESS) target facilities, were purposively sampled by AQCESS project implementation project managers knowledgeable with Kisii and Kilifi. We exclusively targeted HCWs, because AQCESS previously conducted a gender assessment study with service users that provided insights on their experience of DMC [1] and developed a strong rapport with the facility staff.

### 2.3. Methods

Qualitative in-depth interviews of 24 HCWs across the two study sites were conducted.

### 2.4. Interview Process

Ethical approval for this study was obtained from the Aga Khan University, East Africa and National Commission for Science Technology and Innovation research permit NACOSTI/*p*/19/2768 on 3 December 2019. Interviews were held within the facilities at a time convenient to the HCWs. Interviewers were trained by the study Principal Investigator (PI) and familiarized themselves with the interviewer guide (Appendix A). Study interviewers explained the purpose of the study to the participants who were voluntarily asked to consent (Appendix B). The interviewer guide (Appendix A) was used to direct the interview process, and all interviews were audio recorded. After the interview, a debrief statement (Appendix C) was read to each participant who were then given an opportunity to ask questions.

### 2.5. Data Management and Analysis

Data from audio recorders was transcribed verbatim by a qualified transcribing company. Identifiers, such as names, were removed, and all data were transferred to the Monitoring and Evaluation and Research Learning (MERL) unit at the Centre for Excellence in Women and Child Health at Aga Khan University. Transcripts were randomly selected by AL, who read and developed the initial code book using a qualitative data analysis software (NVIVO 12— QSR International (1999) NVivo Qualitative Data Analysis Software [Software]. Available from https://qsrinternational.com/nvivo/nvivo-products/). The code book was used by the Research Assistants (RAs) to code all the remaining transcripts. AL reviewed the coded data and merged the main and sub themes. SW read all the codes and developed the final code book. 

## 3. Findings

Health care workers acknowledged that they observed DMC at times during the delivery of care to pregnant mothers. However, by and large, they do not identify the delivery of this care as intentional nor deliberate. Rather, it appears to be associated with the stress of delivering health care as influenced by various structural, individual, and social cultural factors. The major findings linked to DMC are presented as follows: Infrastructural challenges, including a lack of privacy spaces, equipment, and commodities, hindered the provision of respectful maternity care.Understaffing of health care workers due to health worker shortages at the study facilities caused burnout and fatigue and resulted in frustrations that were reflected in poor attitudes towards clients.Sociocultural differences between health care workers and clients surrounding birthing preferences, expectations, and practices caused communication barriers and tension.Health care workers’ poor attitudes towards clients based on the social and economic status of the clients, including age, disability, and wealth (i.e., poor mothers were more likely to experience DMC; adolescent mothers were likely to be judged harshly by HCWs, as they disapproved of them for becoming pregnant at an early age; mothers living with disabilities were also likely to be discriminated against).

Patterns and examples to support these findings are presented below and are summarized into codes, categories, and themes in Table 1. These specific themes, which emerged from the data analysis, are supported below using various quotes. These findings illustrate and support the emerging themes relating to health care workers’ perspectives of the influences of disrespectful maternity care in Kenya. 

### 3.1. Infrastructural Challenges

#### 3.1.1. Lack of Privacy Spaces

An appropriate and conducive environment with adequate and functional working spaces, working equipment, and commodities are foundational for the proper delivery and maintenance of quality maternal care. All women and newborns are entitled to privacy at the time of delivery. Health care workers from Kilifi illustrate how the lack of privacy spaces created challenges in the delivery of care. They emphasized that privacy spaces are necessary for the provision of respectful and appropriate care. 

“…I see it’s challenging, because of space, our facility not enough. So, we, I have an idea that a mother in labor or delivery, I should offer that privacy …”IDI: Nurse–Kilifi

“…for privacy we need, we’ll need space, which is very important, because we are having very many mothers who actually need service. We have even the doctors, they don’t have anywhere to do their examination, they are using the same couch; so, I think the most important thing to be done here in xxx hospital is space, we need space for that…”IDI: Nurse–Kilifi

#### 3.1.2. Lack of Equipment and Commodities

The availability of essential equipment and commodities is necessary for the provisions of quality maternity care. Infrastructural barriers prevent the appropriate assessment and care of pregnant women, indicating that RMC extends beyond the abilities, actions, and practices of health care workers. In the examples below, nurses recounted how resources were not adequate to meet the needs of mothers seeking care at the facility. 

“…This mother may not know, but for our case we know that we don’t have enough equipment. So it was a challenge because you go ahead using the surgical plate and code gloves. Yes. Because these other ones they have to go through the process of sterilization then to be reused. Contamination all those…”IDI: Nurse–Kisii

“…The first thing, if we can get enough equipment which is needed in the maternity. Secondly, we need like there’s supposed to be a screen when a mother is in the delivery couch even the sub staff who is doing cleaning should not see that mother…”IDI: Nurse–Kisii

“… availability of maternity commodities, at times you may get stock-outs, things like cotton, they may get out of stock, things like gauze ….”IDI: Nursing Officer–Kilifi

These findings point to wider health system gaps in providing sufficient infrastructure, including space, equipment, and commodities that can contribute to DMC. These challenges are not directly amenable to intervention directed at the health workers themselves.

### 3.2. Understaffing

#### 3.2.1. High Patient-Staff Ratios 

Participants in both study sites reported that high patient-staff ratios are one of the greatest barriers in the delivery of respectful maternity care stemming from a shortage of HCWs attending to clients within the health facilities. Health workers have to manage multiple mothers or clients concurrently, and this often leads to them feeling overwhelmed and burnt out and not able to provide continuous, complete, high-quality respectful maternity care.

“…shortage of staff it comes in, suppose the person is alone there are many active cases, now that also contributes, there are burnouts…”IDI: Nurse–Kisii

“…You find another mother wants to deliver, there’s another one who’s bleeding on the other side, the child is having fever on the other side, you see they are competing tasks. You find that if it is a staff who is alone, you find now is overwhelmed and a staff who has been overwhelmed can talk anything…”IDI: Nurse–Kisii

“…This staff is alone, or they are two and they have more clients or many clients in labor and they have to attend to all of them. So that one can bring that tension....”IDI: Nurse–Kilifi

“In many facilities, yes, you can get it that maybe, a staff has 5 mothers, and you are only one staff who is operating in that room. You want to take care of the Post Natal Care department and then laboring those who are still in first stage of labor and you are the same nurse who is supposed to conduct the deliveries, obviously you will be overwhelmed.”IDI: Nurse–Kilifi

#### 3.2.2. Burnout and Fatigue

The examples detailed below, largely from Kisii, describe understaffing as a broad challenge in the public health system and how increases in human resource capacity may create a more motivated workforce to deliver quality maternity care. 

“… now for example in our setup here, you find yourself during the whole night, you are alone, so you handle this case, this case, this case, even you yourself you find that you don’t have that strength let’s say, or you get tired, explaining to that mother that I want us to do this, and this, and this, so because of that you will find that someone doesn’t have even that morale of explaining because you find that they do a lot of work, and they are tired…”IDI: Nurse–Kisii

“…Also to get enough staff because when you are alone even if you provide it reaches a time you get tired and yet you have clients, so it means you will not provide quality services…”IDI: Nurse–Kisii

“…you know sometimes you get a person doing a lot of work, you can say we were two on duty and then we have like five deliveries, or two deliveries, so if somebody comes, at around 5 am in the morning, so you find the nurse is already tired, so if a mother comes let’s say a preemie, she is told push, push then she is not up to what she wants, you’ll find her shouting at her “I wasn’t there when you got pregnant” something like that…”IDI: Nurse–Kisii

The findings indicate that high patient-staff ratios and overwork lead to high levels of stress associated with fatigue and burnout. This stress contributes to DMC, which is associated with a decrease in the tolerance towards mothers’ needs and requests, directly impacting the care provided to the clients.

### 3.3. Sociocultural Influences

#### 3.3.1. Client Preferences for Female Health Care Workers

Health care workers’ accounts illustrate how tensions may arise with mothers and their families displaying a preference for female HCWs to assist them during deliveries due to their cultural and religious beliefs. For example, those of Muslim faith and more traditional mothers and their families often prefer female staff. The issue of client preference frequently poses a dilemma due to staff shortages and existing deployments where the inability to provide the preferred HCW to clients and their families is perceived as DMC.

“…like when you get the Somalis a man could not deliver this mother…”

“…What I think about it, she might need this man, the woman to deliver her but in the real sense the female staff is not there. So, this one as I have said, they need to be health educated, to be, those about the, about the cultures…”IDI-Nurse–Kisii

“…Okay, here in our setup, they most likely women to deliver them…” 

“…Yes, because of the, I can also call the tradition; it’s a traditional belief woman believe that a man cannot look at me…”IDI: Nurse–Kisii

“…With that study, have, okay I only experienced it when I was working in [xxx] county where a mother could come, a pregnant mother, the Somalis and prefer a female nurse to attend to the same as compared to this other communities…”IDI: Nurse–Kilifi

#### 3.3.2. Women’s Cultural Beliefs and Practices Regarding Birthing Practices

The participants from both Kilifi and Kisii also indicated that there are other tensions that may arise in relation to cultural beliefs of clients relating to birthing practices, such as some clients using herbs, a practice that may be discouraged by health workers due to potential perceived harms and uncertainty over the safety of these herbs for mothers and babies. The findings demonstrate the importance of providing a continuity of care and community education relating to cultural influences on birthing practices to bridge the gap while maintaining consideration for individual beliefs.

“…Okay me what I can say about the cultural beliefs what is supposed to be done this people need to be educated. About the cultural beliefs, about the norms, because you cannot encourage them to continue practicing, to continue practicing the, to continue with their norms. So, if it’s delivery, if it’s delivery but not herbs me what I can say in delivery…”IDI: Nurse–Kisii

“…Okay, like in where we are, a negative believe if I may take, like the believe, if you don’t deliver within the time they consider the right time there are those concoctions they normally take and if they take those concoctions, of course there are those, the precautions about the baby so, those are things that we will advise them on the negatives about whatever they take…”IDI: Nurse–Kilifi

#### 3.3.3. Women’s Preference for Home Deliveries

Some clients prefer home birth to facility delivery, and this can be a point of departure with HCWs who encourage skilled attendance at birth consistent with global best practices. Any overt disapproval by HCWs can be misconstrued by mothers and family members as DMC. Female nurses explained how disapproval of cultural practices stemmed from misunderstanding and communication challenges and created situations where mothers were disrespected and treated harshly and in judgmental ways.

“A mother who has been forced to come deliver at the hospital, maybe she is used to squatting, sometimes they do squat, she tells you, I am used to delivering while squatting and maybe you want her to lie on the beds available so she can’t squat. And if you tell her she resists, so it forces you to be harsh for her to lie in the position you can manage to support the baby, something like that.”IDI: Nurse–Kilifi

Mothers’ experiences and/or preferences for home births (aversion to facility deliveries) may create challenges in the delivery of quality care during facility deliveries. Facility deliveries are held to different standards of care compared to home births, and HCWs highlight the importance of community education and sensitization to correct negative cultural perceptions and behaviors while respecting mothers’ autonomy.

“…Mostly, these women they give birth at home, the previous pregnancies, many of the previous babies they give birth at home so if you see this woman has come to the hospital she has maneuvered herself there at home, until she has known that, I won’t deliver by myself at home, so, she comes. So, if she comes there, she is like she has delivered children, so, for a health worker and there is that feeling, so, you expect that woman is used to deliver, is the same woman who is like behaving funny... So, you are trying to advice, but she can’t take that advice … so, she looks like you are treating her as if she has not got some babies before…”IDI: Nurse–Kilifi

“…Yes, there are beliefs in the community, but for example there is a belief, issues of substitution, for example if they eat eggs, they are going to be very fat and there are religions which don’t allow people to deliver in a health facility...”IDI: Nurse–Kisii

### 3.4. Staff Attitudes Towards Marginalized Women

#### 3.4.1. Health Care Workers’ Attitudes Influenced by Women’s Socioeconomic Status

Staff attitudes, to a large extent, are shaped by the social and economic status of the clients, affecting the care they deliver. Poor mothers endure a greater level of disrespectful care, while rich mothers appear to be more demanding and receive a higher quality of care. These staff attitudes in the health care setting may reflect the cultural context wherein HCWs treat clients who appear to have a higher socioeconomic status better, because they are likely to raise concerns over disrespectful care. In contrast, women of lower socioeconomic status may not voice their concerns when treated with disrespect due to the power distance arising from the socioeconomic gap with HCWs and perceived consequences.

“…These low economic mothers they just come the way they are. You may get the mother is coming to deliver, when you see this mother she has not even taken bath, so from there the service provider goes ahead to tell the mother that you have to take a bath before you deliver. There is water there, do what, shower. And yet there are other mothers who are there to deliver. So this mother who is with that information feeling that she has not been handled well…”IDI: Nurse–Kisii

“…Okay, I may not say I have experienced it or I have, but in me I think also because some of this things, the triggers of this things it depends also with the approach of the, of the client because there are times you find there are those learned clients who come very pushy and in the process things may come up to be negative, so you see that somebody who has a high social status. So, but okay usually that could be also possible the low social status…”IDI: Nurse–Kilifi

“…So, it is true, some people judge mothers they will give very good and quality care to mothers who maybe looked maybe look to be they are employed teachers or those who look to be economically…”IDI: Nurse–Kilifi

“…with this issue of classifying mothers depending on their social standards of life is an aspect that is there. So, a mother comes and the other one is, belongs to the higher standard of, social standards of life, then the preference of the, the health provider, health provider, service provider will attend to the one who is well off than the other one…”IDI: Nurse–Kilifi

#### 3.4.2. Disability

Despite these contrasting observations, the HCW experiences demonstrate an implicit bias toward women living with disabilities that ultimately direct their care. A HCW reported that mothers living with disabilities may face premature biases that influence the care that they receive. 

“…Okay these disabled mothers, before the service provider, the problem we have are that we judge this mother before we get the information. Like now you may get that the service provider thinks that this mother cannot deliver normally. So before you get the whole information, you just tell the mother no you go to the facility that has theater. So you have informed the mother what is not there and maybe you get this mother already she has other two babies she has delivered normally but you see she is unable you just judge it and you decide what to happen…”IDI: Nurse–Kisii

In contrast, these observations from nurses in Kilifi and Kisii indicate that women living with disabilities may receive higher standards of care.

“…No, I think they are given the best service those that I have witnessed. Because there is that empathy you feel for that…”IDI: Nurse–Kilifi

“…this disabled mother or this the poor mother the rich one, all of them there’s that what we call equality and there’s that what we call first come first served. You can’t abandon the, or you can’t ignore the poor mother or the disabled and you run for the woman who has a car. So, to me I don’t see…”IDI: Nurse–Kisii

#### 3.4.3. HCW Attitudes towards Young Adolescent Pregnant Women

Our data also showed that health workers may have their own internal algorithms that dictate their attitudes towards patients—in particular, their poor attitudes toward pregnant teenagers stemming from social disapproval of premarital sex and discontinued education. This stigma may lead them to provide DMC to adolescent girls who become pregnant. This is so regardless of the factors that may have led them to become pregnant, including experiences of gender-based violence.

“…Youth, the adolescents…”

“…They are young, and they are the most at risk, so if you don’t handle well, they see it as lack of respect…” 

“…Because they don’t even know how to express themselves, and also the attitude we have, that they are so young they are supposed to be maybe in school, they engaged in premarital sex at a tender age…”IDI: Nurse–Kisii

“…An adolescent mother, I mean, girl who has gotten pregnancy and at the age of where she is supposed to be still in school and whoever be is in the ANC clinic is somebody who is aged at the age of her mother, sometimes they start some kind of stigma like: why are you here? Why do ask? Why have you come to this place? You know there is that attitude, it is the attitude for the staff, which is against the professional code of ethics, but it is instilled in them. So, it is like she is going to ask more questions that are relating to very specific personal needs apart from those which are generalized as per the condition this young mother has come for, simply because of the stigma and attitude that mother, I mean that health worker has towards that client. Yes, it is there…”IDI: Nurse–Kilifi

“…Teenagers, some, there are some who can treat them like it’s not me who got you pregnant, and such things, you are a student, but it is not good. You just take them because it has happened. You can’t throw them away because next time she can even tell the others there; there you are insulted badly, or you are beaten. So, we just treat them but in most of the cases, teenagers, they are, mostly they are mistreated. …”IDI: Nurse–Kisii

The inexperience of adolescent mothers was also highlighted as a barrier to the delivery of quality care by respondents. The examples below illustrate communication challenges and how adolescent mothers’ lack of information may create strain between health care workers and the young mothers. Adolescents may not confidently express themselves in the health care encounters due to the power dynamic with older HCWs who wield a lot of authority in a largely paternalistic setting.

“But sometimes you see the adolescents dealing with them is hard, you want to do a vaginal exam and she doesn’t want to open the thighs, it’s hard, sometime you find that you talk in a harsh way, but later she will comply and you do the vaginal exam. So, maybe sometimes you may try to communicate with the youth and maybe it will come out in a harsh way because you want to do that vaginal exam to help her to know the progress of labor.”IDI: Nurse–Kilifi

“…most of the young women, we know some of them are shy in nature, they not easy to give out information so may be the service provider may feel tired for asking so many questions that the client is not responding so they may be tempted to be harsh to them.”IDI: Nurse–Kilifi

“…Sometimes, it can be age, because this one is young and the other one is older. And another thing its communication, because first you must communicate to me what you want to do for me and then, so once you have communicated back to me, am ready maybe you are going to do vaginal examination, so that one is “sleep there” I think its communication and then age difference…”IDI: Nurse–Kisii

These narratives of the health care workers in this study illustrate some of the negative attitudes that health workers may have towards clients accessing maternal services. These vignettes also illustrate some of the attitudes and the underlying influences associated with the delivery of maternal care in Kenya.

### 3.5. Summary of Findings

The findings that emerge from the analysis above identify four different themes that address barriers that may inhibit the use of maternity facility-based care. These themes can be categorized into two main views on maternity care, including the provision of sufficient and quality resources and the experiences of the delivery of care. Figure 2 presents a conceptual framework that summarizes these perceptions and connected themes as they impact the delivery of maternal care. 

## 4. Discussion

Using qualitative, in-depth interviews with HCWs, our study identified factors from the HCWs’ perspective that may contribute to the poor treatment of pregnant women seeking antenatal care (ANC), delivery, and postnatal care (PNC) services at facilities in rural Kisii and Kilifi. The findings are intended to promote our understanding of factors contributing to DMC beyond reports by those that receive care, as well as to provide a more nuanced explanation on the factors that encourage HCWs to mistreat pregnant women under their care. 

Overall, our study showed that health care workers acknowledge that DMC is experienced by mothers seeking maternal care services in Kenya, corroborating findings from previous studies from Kenya and other regions in sub-Saharan Africa that examined the perceptions of service users [1,2,3,4,5]. Moreover, the findings across studies of HCWs and service users are markedly consistent in identifying barriers to respectful care and point towards interventions that address health infrastructure, staffing, sociocultural beliefs, and, also, promote positive behavioral changes towards age-, gender-, disability-inclusive practices and the destigmatization of marginalized women, including pregnant teenagers and women living with disabilities. 

Similar to our study findings, the poor infrastructure, such as lack of privacy spaces and the lack of resources and commodities, contributing to DMC, have been reported in other studies [13,14]. The poor design and size of health facilities often violate privacy, as women are exposed to other patients, their families, and providers [14]. The increase in the number of patients due to the recently implemented Free Maternity Policy (FMP) has also added pressure on existing facilities [19,20]. Consistent with this, we also found that understaffing creates a work environment conducive to DMC. Feelings of exhaustion, fatigue, and burnout, which increase stress and reduce the level of tolerance of HCWs, may lead to caregivers being short with mothers and acting rudely towards those in need of care. Similar findings have been reported whereby frustration stems from stressful work environments, resulting in HCWs transferring their negative feelings onto the mothers [14,21]. Further, the evidence shows that a poor working environment drives poor interpersonal communication among HCWs providing maternity care [21]. As such, health care organizational environments need to be improved and better-staffed to prevent burnout of HCWs. These strategies must be reinforced by wider health system supports in relation to adequate staffing, resources, and commodities to promote respectful maternity care.

While the health system influences on DMC are a structural reality, HCWs also admit to exhibiting judgmental and stigmatizing attitudes that breed discriminatory actions towards mothers that negatively impact care-seeking practices. This finding supports previous research that has shown that women fear discrimination during facility-based delivery based on poverty, age, and culture [15]. The stigmatizing attitudes and poor interpersonal communication skills are within the sphere of control of the health workers and are amenable to intervention; therefore, strategies should target actions that incentivize behavior changes to reduce implicit stigma [9]. Stress reduction strategies, as well as training, education, and supervision, are required to better address the internal algorithms that health workers may have that manifest in the form of judgmental attitudes towards mothers [16,21].

The findings indicate that HCWs have good awareness of the influences of DMC, which is consistent with users’ experiences of DMC reported in our previous study in the two counties [1]. Our findings that adolescents and women of lower socioeconomic status are more likely to experience DMC in the hands of health care workers are also consistent with findings from Nigeria, where young mothers experienced judgmental attitudes because of their perceived lack of experience [14]. These findings suggest that training geared toward education and behavior change strategies may be an effective way to improve respectful attitudes and care delivery among HCWs [9].

### 4.1. Proposed Recommendations

Factors that influence DMC from both the service users and HCWs are structural, social cultural, and individual. Respectful care should be part of HCWs’ preservice and in-service training curriculum. Further, since the variables that influence DMC may reinforce each other, an intersectional approach would be a useful approach to allow for a careful analysis of the connections between barriers to DMC from both users’ and HCWs’ perspectives and possibly inform strategies to help HCWs to provide high-quality maternal health care services. This implies that addressing the barriers that amplify DMC may require the contributions of different stakeholders at different levels of policy-making. The intersectional approach will help to identify where inequities occur, the structural and policy features that contribute to these inequities, and the social and cultural factors that determine peoples’ behaviors and choices. In this approach, interventions seek to address how intersecting social categories, of both the HCWs and service users, such as social class, age, disability, religion, attitudes, beliefs, and norms, and infrastructure challenges contribute to DMC. 

We emphasize that using an intersectional approach will embrace the multiple viewpoints of stakeholders. As such, these could help in developing all-inclusive and sustainable policies geared towards combatting healthcare workers’ attitudes and behaviors that disadvantage and compromise women’s quality of health care based on social categories. For example, this may help develop guidelines of care that are suitable for all pregnant women regardless of their situation. As well, tailored training for HCWs and with the community members could help to eliminate some of the sociocultural barriers. In addition, defeminization of the birthing process by engaging more men at the grassroots level to encourage women to accept the attendance of male nurses at maternity facilities could help in diluting myths around gender and birthing. Moreover, health facilities should have a mix of male and female health care workers in health facilities to encourage the attendance of hard-to-engage facility users like mothers whose first preference, as research has shown, is deliver from home with the help of TBAs. Lastly, intentional strategies are needed to effect behavior changes among health care workers directed at addressing the stigma and discrimination of mothers due to socioeconomic standing.

### 4.2. Proposed Research

This current study of HCWs, combined with the previous study on service users, provides a platform upon which future, large national surveys can be developed to provide a wider understanding of RMC in both urban and rural areas across Kenya and sub-Saharan Africa. For instance, a cross-sectional national survey to examine “factors that contribute to negative staff attitudes towards low social-economic women during delivery” and “what HCWs are likely to provide DMC” will increase our understanding on why poor women are marginalized. The findings will also provide the scaffolding for intervention studies of changes and alterations to various barriers to RMC and to assess the efficacy and effectiveness of these changes on health outcomes of mothers and babies. The two studies offer rich and corroborating data upon which actions and consultative panel research can be developed to provide insights into potential interventions. For example, a joint consultative panel study with service users, including traditional birth attendants (TBAs), as well as facility HCWs, could deliberate on whether herbal treatments that mothers use are unsafe or unhealthy and under what circumstances should these herbs be used. Finally, these findings can also be used to develop key informant tools for policy-makers to inform, as well as advise on, national and local policy, as well as nursing and medical school curriculum.

## 5. Limitations and Strengths

This study was conducted at health facilities in two rural counties where the AQCESS project was implemented in Kenya and may not reflect the perspectives of HCWs in the country as a whole. The study participants were recruited by the AQCESS project managers who were familiar with them, and considering the sensitivity of the topic, they may have underreported some of their views and observations. However, a study such as this focusing on such a sensitive topic was only possible due to the strong and trusted connections cultivated through the AQCESS project. This study provided rich data that not only deepened our understanding but closed some of the existing knowledge gaps can be used to inform policy and practice in rural Kenya. A major strength of this study was its direct focus on HCWs’ perspectives to complement most studies that focused on user experiences. The rich data in this study based on the perspectives of HCWs at the frontline in the provision of maternity care is important for illuminating and interpreting DMC, an important aspect of developing relevant strategies to improve care.

## 6. Conclusions

This study attempted to complement the existing body of research on the recipients of maternal care by providing critical evidence on health care workers’ perspectives of the influences of DMC. There is acknowledgement by health care workers that DMC occurs, and there is a remarkable alignment as to their perspective of the causes with that of those receiving care. These findings promote a clearer understanding of the health system supply side factors that act as impediments for women utilizing maternity health care facilities, thereby contributing to poorer maternal and neonatal health outcomes. 

## Figures and Tables

**Figure 1 ijerph-17-08218-f001:**
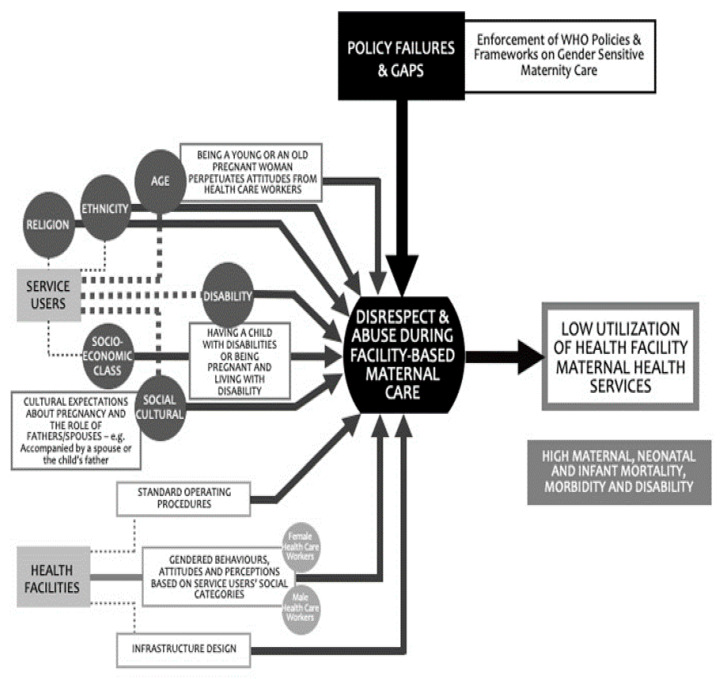
A summary of factors that influence disrespectful maternity care during facility care in sub-Saharan Africa (Lusambili et al., 2020) [1].

**Figure 2 ijerph-17-08218-f002:**
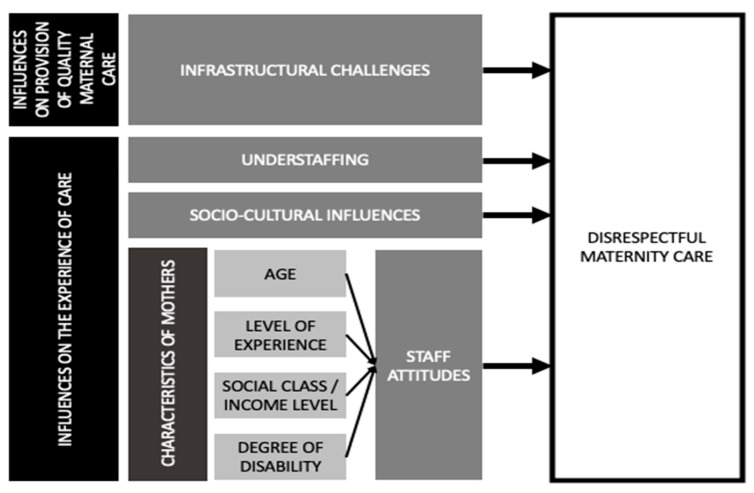
Conceptual framework to understand the influences of health care workers’ (HCWs’) disrespectful maternity care.

**Table 1 ijerph-17-08218-t001:** Codes, categories, and themes of health care workers’ experiences with disrespectful maternity care.

Codes	Categories	Themes
We have fewer maternity equipment. We have dysfunctional equipment. We have fewer beds. We lack basic commodities. We lack private spaces for our doctors. We lack private spaces for our clients.	1.1 Inadequate infrastructure 1.2 Lack of equipment 1.3 Stock out of commodities	1. Infrastructural Challenges
We are the only two on the night shift. I am alone on the shift. I cannot manage alone—the patients are many. I have many clients during a shift. I get exhausted. It is difficult.	2.1 Fatigue and Burnout	2. Understaffing
Muslim/Tribal beliefs about female/male nurses. Home deliveries versus facility-based deliveries Cultural beliefs relating to birthing practices. Use of herbs during delivery.	3.1 Preference for female health care worker due to culture 3.2 Use of herbs during pregnancy 3.3 Differences in cultural beliefs between clients and health care workers 3.4 Client preference	3. Sociocultural Influences
Being a young adolescent mother. Being a mother living with disabilities. Being poor/rich—poor mothers are ignored.	4.1 Stigma and discrimination based on socioeconomic characteristics of mothers	4. Staff attitudes towards marginalized women

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
