# Peer review of "Health Care Workers’ Perspectives of the Influences of Disrespectful Maternity Care in Rural Kenya"

_ijerph, 2020, doi:10.3390/ijerph17218218_

Round 1
Reviewer 1 Report
The authors did an original study titled “Health care workers’ perspectives of the influences
of disrespectful maternity care in rural Kenya: The topic is very interesting, however there are minor concerns needed be addressed.
Abstract:
Line 16: or has studied disrespectful maternity care (DMC)…..rephrase or remove “or”
Line 17: health care workers (HCWs) Not (HWCs)
Introduction: I will suggest a more extensive literature review is carried out not just to demonstrate your understanding of the subject matter but justifying your study and setting your research question.
Line 45: “Disrespectful maternity care then would be the opposite or absence of this care” This sentence is misleading and a bit confusing. I suggest you rephrase the sentence by remove “opposite” and clarify the care you are referring to.
Line 63 and Line 65: Use square brackets for uniformity.
Line 404: “stigmatization by HCWs of marginalized women including pregnant teenagers and women living with disabilities” This sentence is somewhat confusing I suggest you rephrase.
Author Response
REVIEWER 1
Comments and Suggestions for Authors
The authors did an original study titled “Health care workers’ perspectives of the influences of disrespectful maternity care in rural Kenya: The topic is very interesting, however there are minor concerns needed be addressed.
Abstract:
• Line 16: or has studied disrespectful maternity care (DMC)…..rephrase or remove “or”
Thank you. This has been addressed.
• Line 17: health care workers (HCWs) Not (HWCs)
Thank you. This has been addressed.
• Introduction: I will suggest a more extensive literature review is carried out not just to demonstrate your understanding of the subject matter but justifying your study and setting your research question.
Thank you.
Literature from HICs has been added.
Justification of the study has also been added.
• Line 45: “Disrespectful maternity care then would be the opposite or absence of this care” This sentence is misleading and a bit confusing. I suggest you rephrase the sentence by remove “opposite” and clarify the care you are referring to.
Thank you. This has been addressed.
• Line 63 and Line 65: Use square brackets for uniformity.
Thank you. This has been addressed.
• Line 404: “stigmatization by HCWs of marginalized women including pregnant teenagers and women living with disabilities” This sentence is somewhat confusing I suggest you rephrase.
We have rephrased this sentence.

Reviewer 2 Report
The study is strong. The interview is well constructed. The limitations are well described.
The link between insufficient supplies (as compared to lack of private spaces) and disrespectful maternity care (DMC) is a bit tenuous. One of the three quotations under insufficient supplies references lack of a privacy screen, which makes more sense as directly related to DMC.
Regarding differences in cultural beliefs as a cause of burnout, can you explain whether the herbal treatments that HCWs disapprove of that mothers apparently seek are actually unsafe or unhealthy, or are they neutral with respect to the health and safety of mother and infant? It could be that here HCW education might be needed if the herbals are not harmful.
Regarding staff attitudes toward marginalized women, this is a topic that may need more explanation. Is this a culturally embedded attitude (i.e. that income level predicts attitudes throughout the community) or is it specific to the healthcare setting? Is it really related to hygiene or what is the aspect that HCWs find disturbing about low income mothers? I think readers might like to understand more here about the cultural context for this domain, if possible.
Regarding adolescent mothers who do not want a vaginal exam, was there any thought at any time that the teen may have been raped, rather than having been “promiscuous” which appears to be the attitude of the HCW?
Since there are a spectrum of attitudes, for example the attitudes about disabled women vary, can you explain if certain HCWs seem to have more bias or propensity to provide unintentionally DMC or were most interviewees/participants’ responses suggest of a mixed picture for most?
4.1- The “intersectional approach” is a bit confusing. If your recommendations follow from the diagrams it would seem there are two clear pathways to eliminating DMC – facility change and HCW attitude change. These would follow two different pathways since one involves funding and structural changes and the other involves education and health care training. Perhaps this is what you meant but this section is unclear. Another analogous recommendation with respect to HCW training is for medical staff to partner with the HCWs to create their own trainings since the level of insight is high with respect to causes of DMC.
Author Response
REVIEWER 2
Comments and Suggestions for Authors
• The study is strong. The interview is well constructed. The limitations are well described.
Thank you.
• The link between insufficient supplies (as compared to lack of private spaces) and disrespectful maternity care (DMC) is a bit tenuous. One of the three quotations under insufficient supplies references lack of a privacy screen, which makes more sense as directly related to DMC.
Thank you
• Regarding differences in cultural beliefs as a cause of burnout, can you explain whether the herbal treatments that HCWs disapprove of that mothers apparently seek are actually unsafe or unhealthy, or are they neutral with respect to the health and safety of mother and infant? It could be that here HCW education might be needed if the herbals are not harmful.
Thank you. You raise an important point. Based on the researcher’s knowledge and few interviews conducted with HCWs in two rural settings in Kenya, - it is not possible to authoritatively assert that the herbal treatments that HCWs disapprove of that mothers seek/use are unsafe or unhealthy.
There is need for more research to explore this issue from both the service users and providers’ perspectives. We have therefore recommended this in the future research consideration section.
• Regarding staff attitudes toward marginalized women, this is a topic that may need more explanation. Is this a culturally embedded attitude (i.e. that income level predicts attitudes throughout the community) or is it specific to the healthcare setting? Is it really related to hygiene or what is the aspect that HCWs find disturbing about low income mothers? I think readers might like to understand more here about the cultural context for this domain, if possible.
Thank you.
1. The issues you have raised are real, not only in Kenya where this study was conducted, but also in High Income Countries where research illustrates that low income mothers for instance- ethnic women/or and/migrant/refugee women are disrespectfully treated by HCWs. [ I have added this in the literature section].
2. In this study, as we have reported, HCWs observed that women who are advised to shower may feel demeaned and think that they are disrespected even when the HCWs mean well.
See our quotes as follows:
“…These low economic mothers they just come the way they are. You may get the mother is coming to deliver, when you see this mother she has not even taken bath, so from there the service provider goes ahead to tell the mother that you have to take a bath before you deliver. There is water there, do what, shower. And yet there are other mothers who are there to deliver. So this mother who is with that information feeling that she has not been handled well…”
IDI: Nurse–
3. On the other hand, as the HCW explains in the following quote, educated women are likely to know their rights as opposed to uneducated women.
it depends also with the approach of the, of the client because there are times you find there are those learned clients who come very pushy and in the process things may come up to be negative, so you see that somebody who has a high social status. So, but okay usually that could be also possible the low social status…”
IDI: Nurse–Kilifi
To make context based explanation why this is happening, we have suggested further exploration of this aspect in future research. A cross sectional survey of HCWs and service users to explore “Factors that contribute to negative staff attitudes to low social-economic women during delivery” will promote our understanding why this is happening.
• Regarding adolescent mothers who do not want a vaginal exam, was there any thought at any time that the teen may have been raped, rather than having been “promiscuous” which appears to be the attitude of the HCW?
The thought of the adolescent mothers having been raped did not feature in the study. Generally, there is a high teenage pregnancy in Kilifi and Kisii where this study was conducted. In part, the educational opportunities favor a boy child, girls marry early, and as reported in the ‘study context’ section of the paper, teenage pregnancy is extremely high in the two study sites.
• Since there are a spectrum of attitudes, for example the attitudes about disabled women vary, can you explain if certain HCWs seem to have more bias or propensity to provide unintentionally DMC or were most interviewees/participants’ responses suggest of a mixed picture for most?
Thank you. In this study, as the quotes show, responses show that there was a mixed picture. In addition, we had few (6) male HCWs recruited in the study. In our sequel paper one (https://pubmed.ncbi.nlm.nih.gov/31910210/), our study findings showed that female HCWs practiced DMC. We have recommended this for further research.
• 4.1- The “intersectional approach” is a bit confusing. If your recommendations follow from the diagrams it would seem there are two clear pathways to eliminating DMC – facility change and HCW attitude change. These would follow two different pathways since one involves funding and structural changes and the other involves education and health care training. Perhaps this is what you meant but this section is unclear.
Thank you for this comment. We feel that it is both but that they are not independent from one another.
• Another analogous recommendation with respect to HCW training is for medical staff ¬to partner with the HCWs to create their own trainings since the level of insight is high with respect to causes of DMC.
The writing team has reviewed the recommendation section.

Reviewer 3 Report
The article is interesting and well put together. It addresses a relevant theme consistent with the journal scope. It is one of the rare cases in which i tis almost ready for publication it that is the journal’s choice. However the references section need a profound revision as they do not follow at all the journal criteria for the correct inclusion of references.
Author Response
REVIEWER 3
Comments and Suggestions for Authors
The article is interesting and well put together. It addresses a relevant theme consistent with the journal scope. It is one of the rare cases in which i tis almost ready for publication it that is the journal’s choice. However the references section need a profound revision as they do not follow at all the journal criteria for the correct inclusion of references.
Thank you. We have revised the references to follow the journal criteria
